# Advances in the Formation and Control Methods of Undesirable Flavors in Fish

**DOI:** 10.3390/foods11162504

**Published:** 2022-08-19

**Authors:** Tianle Wu, Meiqian Wang, Peng Wang, Honglei Tian, Ping Zhan

**Affiliations:** 1College of Food Engineering and Nutritional Science, Shaanxi Normal University, Xi’an 710119, China; 2The Engineering Research Center for High-Valued Utilization of Fruit Resources in Western China, Ministry of Education, Shaanxi Normal University, Xi’an 710119, China

**Keywords:** undesirable flavor, lipid oxidation, microorganism, deodorization, natural active substances

## Abstract

Undesirable flavor formation in fish is a dynamic biological process, decreasing the overall flavor quality of fish products and impeding the sale of fresh fish. This review extensively summarizes chemical compounds contributing to undesirable flavors and their sources or formation. Specifically, hexanal, heptanal, nonanal, 1−octen−3−ol, 1−penten−3−ol, (E,E)−2,4−heptadienal, (E,E)−2,4−decadienal, trimethylamine, dimethyl sulfide, 2−methyl−butanol, etc., are characteristic compounds causing off−odors. These volatile compounds are mainly generated via enzymatic reactions, lipid autoxidation, environmentally derived reactions, and microbial actions. A brief description of progress in existing deodorization methods for controlling undesirable flavors in fish, e.g., proper fermenting, defatting, appropriate use of food additives, and packaging, is also presented. Lastly, we propose a developmental method regarding the multifunctional natural active substances made available during fish processing or packaging, which hold great potential in controlling undesirable flavors in fish due to their safety and efficiency in deodorization.

## 1. Introduction

Flavor is one of the most important palatable characteristics of food, significantly affecting food quality and consumer acceptance. Fish have long been considered an excellent source of high−quality protein. However, some unpleasant odors in fish caused by bacterial growth, the environment, processing methods, and storage conditions have still not been fundamentally resolved, which restricts fish processing in foodstuff. Therefore, promoting the quality of flavor and utilizing fish resources effectively have become one of the hottest research topics in fish processing.

Fish in general are classified into two main types, namely, fresh− and saltwater fish. The undesirable flavors of freshwater fish, such as earthy/muddy, fishy, and grassy, come from multiple sources [1]. Among these, the earthy odor is primarily caused by the metabolites of the inhabitant microorganism flora, i.e., mainly cyanobacteria (e.g., Anabaena, Lyngbya, Microcystis, and Skeletonema) [2] and actinomycetes (e.g., Streptomyces) [3]. Saltwater fish often have less intense undesirable odors than freshwater fish [4], and normally, saltwater fish tend to release “sea breeze−like” odors [5]. The differences in undesirable flavor profiles between freshwater and saltwater fish are the result of different volatile odor compositions; specifically, freshwater fish have more aldehydes, contributing to stronger fishy and grassy aromas, and have a stronger earthy odor, caused by geosmin (GSM), when compared to saltwater fish [5,6].

Cultured fish for human consumption contain richer lipids within their muscle tissue than fish living in the wild [7]. Experimental results showed that the relative concentration of lipid−derived volatile compounds was significantly higher (*p* < 0.05) in aquacultured fish samples as compared to wild samples [8], from which it may be concluded that oily fish often have more volatile oxides than lean fish. Sensory differences were also measured in a comparison of wild and cultured fish; assessors described wild gilthead sea bream as having ‘‘a more pleasant taste’’ and ‘‘a firmer texture’’, while the cultured group was thought of as having ‘‘poor taste’’, indicating the superiority of the wild fish [9,10].

Consumers demand commercial fish products with no unpleasant aroma or taste. An improved understanding of the compositions and formation of undesirable flavors helps to develop the flavor quality of fish products. We thus review the unpleasant components and their various sources, including enzymatic reactions, lipid autoxidation, environmentally derived reactions, and microbial actions. Meanwhile, most deodorization methods are summarized and evaluated in this review, and the use of natural deodorization methods can be highly effective while remaining environmentally friendly.

## 2. Volatile Compounds Contributing to Undesirable Flavors in Fish

Strong undesirable flavors are the most significant problem encountered in aquaculture, causing the dissatisfaction of consumers and a reduction in the market value of the product. Shi et al. reported that the primary volatile compounds in fish are alcohols and carbonyl compounds (the total relative contents are above 90%) [11]. Among these compounds, 2−−/3−methylbutanal, hexanal, heptanal, octanal, nonanal, (E,E)−2,4−heptadienal, (E,Z)−2,6−nonadienal, (E,E)−2,4−decadienal, (Z)−4−heptenal, (E)−2−octenal, (E)−2−nonenal, 1−penten−3−ol, 1−octen−3−ol, acetic acid, butanoic acid, (E,E)−3,5−octadien−2−one, (Z)−1,5−octadien−3−one, etc., play an important role in the characteristic flavor of fish, and most of them have been identified in many types of fish [5,8,12]. On the one hand, the same volatile compound can give off several flavors and odors: for example, 1−octen−3−ol not only has mushroom−like and strong plant−like flavors but also has a metallic−like flavor; 2,4−decadienal gives off orange, fresh, fatty, or green aromas [13]. On the other hand, a certain flavor is composed of a complex mixture of volatiles; for example, the fishy or rancid odor of fish products primarily consists of hexanal, (E,E)−2,4−decadienal, (E,E)−2,4−heptadienal, heptanal, etc. [14]. In general, a few relatively low molecular weight aldehydes mixed together are responsible for the pungent fishy smell of fish [8,11].

Previous research has shown that in a recirculating aquaculture system (RAS), earthy/musty odors originating from geosmin (GSM) and 2−methylisoborneol (MIB) are the most perceivable undesirable flavors in cultured freshwater fish [6,15]. It has also been reported that MIB contributes to the musty odor, while GSM is responsible for the earthy odor [16,17]. Moreover, GSM and/or MIB co−existing with aldehydes and alcohols such as hexanal and 1−octen−3−ol notably intensify the musty off−flavor in catfish fillet [18]. β−Cyclocitral, a characteristic undesirable flavor compound in fish providing tobacco/smoky/moldy smells [19], also contributes to off−flavors to fish in ponds [20]. Existing findings suggest that fish directly absorb these chemical compounds from ambient water and rapidly store them within their fatty tissue [21].

Characteristic sulfur and nitrogen derivatives are also regarded as volatile off−odor compounds, both of which are usually related to the deterioration of seafood. Due to low threshold values, the compounds can influence the overall aroma even in very low proportions [22]. Dimethyl sulfide (DMS) is considered the typical spoilage marker of sulfur−containing compounds and has been detected at low concentrations in some freshwater fish species [23]. Similarly, trimethylamine (TMA) is also regarded as a fish microbial spoilage marker and is used as a potential indicator of fish freshness. TMA comes from the reduction of trimethylamine−oxide (TMAO) [24], which occurs in saltwater fish and plays a significant role in keeping pH and osmoregulation stable, and the formation of TMA is accompanied by ammonia−like and fish−house−like odors [25]. It has been suggested that TMA in combination with DMS significantly contributes to saltwater fish odors, producing a fishy and seafood flavor that is much stronger than that of TMA alone [4].

Freshly harvested saltwater fish often have pleasant seaweedy, sweet, and faint green aromas. However, neutral to acid odors develop during storage, culminating in an overall pungent odor of fish and resulting in lower sensory grades (Figure 1) [26]. Triqui et al. [26,27] assumed that the concomitant elevation of concentrations of (Z)−4−heptenal, (Z)−1,5−octadien−3−one, and methional correlated with the development of an overall fishy odor in sardine during ice storage. Likewise, a study revealed that after storage of the raw material, the OAVs (odor activity value: a ratio of concentration to odor threshold) of (Z,Z)−3,6−nonadienal and (Z)−3−hexenal were significantly enhanced, which are responsible for the fatty and fishy off−flavors of boiled trout [28]. Moreover, it is noteworthy that both cultured and wild fish showed complex volatile profiles throughout the entire storage period; for example, rancid, putrid, sulfurous, and ammonia−like odors in fish are attributed to volatile odor compounds such as TMA, dimethyl disulfide, dimethyl trisulfide, piperidine, 1−penten−3−ol, 3−methyl−1−butanol, methanethiol, and acetic acid [24].

## 3. The Formation of Post−Harvest Undesirable Flavors in Fish

### 3.1. Lipid Oxidation

Many compounds causing undesirable flavors may originate from the process of lipid oxidation. Lipid oxidation is a major cause of poor quality and is also responsible for causing the development of fishy odors in fish, as well as undesirable textural changes through the interaction of protein and lipid oxidation products [29,30]. Phospholipids containing polyunsaturated fatty acids within the cell membrane are closely related to lipid oxidation in fish since they have a high degree of unsaturation and large surface area that are susceptible to oxidation [31]. Previous analyses have shown that the incidence rate of oxidation depends not only on the amount of lipids in the sample but also on the oxidative conditions, the enzymatic activity of the lipoxygenases, and the abundance of the antioxidant compounds present [32]. Enzymic involvement is necessary for the generation of lipid−derived volatiles in fresh fish; however, research indicates that lipid peroxidation in nonliving fish tissue is initiated nonenzymatically, primarily by heme−protein autoxidation [33,34]. In fact, lipid oxidation in the fish muscle is induced by several catalysts, including hemoglobin, iron, and lipoxygenases; furthermore, lipoxygenases result in severe fishy odors, while hemoglobin can be affiliated with a strong oxidized oil odor [35,36].

Oxidative enzymatic reactions and the autoxidation of lipids contribute to fresh fish with green, plant−like, metallic, and fishy aromas to a significant degree [37]. The main volatiles derived from lipids are reported in Table 1. Notably, biochemical reactions of polyunsaturated fatty acids (PUFAs) with lipoxygenases and hydroperoxide lyases produce unsaturated carbonyls and alcohols, such as 5−, 6−, 8−, 9−, and 11−carbon alcohols and carbonyls (e.g., eicosapentaenoic acid, shown in Figure 2) [12,13]. Meanwhile, autoxidation of PUFAs produces other types of unsaturated carbonyls, such as 6−, 7−, 8−, and 10−carbon carbonyls. In addition, enzymatic and nonenzymatic oxidation can occur simultaneously during cooking [38], and the most polyunsaturated of all of these acyl groups shows the highest tendency to undergo autoxidation. Previously, the fish muscle was covered by fish skin and was less susceptible to oxidation; therefore, the muscle released a minimal fishy flavor compared to the fish’s skin when perceived by smell [39]. Furthermore, the rate of lipid oxidation of the yellowtail dark muscle was faster than that of the ordinary muscle. The total lipid hydroperoxide content and thiobarbituric acid−reactive substances (TBARS) of the dark muscle were significantly higher than those of the ordinary muscle after 2 days of ice storage [40]. In addition to the characteristics of the fish itself, some physical factors also affect the degree of oxidation. For example, heme proteins, which are well−known prooxidants, can be further activated during the pH−shift process [34]. The intensity of heat treatment plays an important role in the extent of oxidation in cooked meat, with higher temperatures generally making hidden volatiles identifiable and accelerating retro−aldol condensation. As a result, more (Z)−4−heptenal and acetaldehyde are formed from (E,Z)−2,6−nonadienal, and (Z)−2−pentenal and acetaldehyde have the potential to be formed from (E,Z)−2,4−heptadienal [41].

### 3.2. Microbial Metabolites

Fish is a high−protein product that is susceptible to the proteolytic activity of microorganisms. Some related studies have reported that the microbial spoilage of harvested fish accounted for the loss of approximately 10% of fish catches worldwide [47,48]. Furthermore, spoilage from microorganisms produces metabolites responsible for various unpleasant undesirable flavors, leading to the eventual sensory rejection of fish products. The organisms causing the highest spoilage potential in specific products or storage conditions are named specific spoilage organisms (SSOs). *Shewanella putrefaciens* and *Pseudomonas* spp. are generally recognized as the specific spoilage bacteria of fresh fish, regardless of the origin of the fish [49]. *Pseudomonas*, *Shewanella*, *Lactobacillus,* and *Carnobacterium* species, identified by 16S rRNA gene sequencing analysis, were proved to be SSOs of gilt−head sea bream at various temperatures and atmospheric conditions [50]. Meanwhile, *Carnobacterium*, *Serratia*, *Shewanella,* and *Yersinia* were shown to be the dominate species in horse mackerel fillets at the time of sensory rejection [51]. Additionally, *Photobacterium phosphoreum* and *Psychrobacter* are the most common SSOs reported in fish products [52,53,54].

In general, different growth substrates can affect the growth rates of microorganisms, such as different fish species or the fish flesh in different process conditions. For example, psychrotolerant Gram−negative bacteria (e.g., *Pseudomonas* spp. and *Shewanella* spp.) are reported to grow in chilled fish, whereas fermentative bacteria (e.g., *Vibrionaceae*) prefer to multiply in unpreserved fish [55]. In addition, different water temperatures, seasonal variation, geographical location, surrounding gaseous composition, and processing might also have complex effects on the initial microbiota [56]. Specific spoilage microflorae dominating in fresh fish meat during storage under different gas atmospheres are shown in Table 2, which potentially determine the composition of the primary microflora [57]. Meanwhile, SSO populations were found to be significantly higher in retail−derived catfish in comparison to lab−filleted catfish tissue [58], and some chemical indicators of spoilage, such as the TMA value, in ungutted sea bass increased slowly, while the TMA values of gutted samples were much higher [59], suggesting that mishandling during processing is a major reason for rapid fish tissue spoilage.

Volatile compounds are associated with the metabolic activities of particular microbial groups. *Pseudomonas* spp. produce large amounts of volatile alcohols, ketones, esters, and sulfides (except H_2_S), whose typical descriptions are fruity, rotten, and sulfhydryl flavors in iced fish [49]. *Shewanella* spp. release intense undesirable odors as well, producing H_2_S and biogenic amines, as well as reducing TMAO to TMA, and even show proteolytic activity at low temperatures [60,61]. Moreover, *Aeromonas* spp., psychrotolerant Enterobacteriaceae, *Photobacterium phosphoreum,* and *Vibrionaceae* were all able to utilize TMAO in order to form TMA, resulting in off−odors [22,55]. *Serratia fonticola* and *Serratia liquefaciens*, *Aeromonas*, *Acinetobacter,* and some *Pseudomonas* spp. are known to produce histamine [62]. It is because microorganisms can utilize different precursor compounds that volatile metabolites are subsequently generated. As shown in Table 3, ethanol, organic acids, and esters are produced primarily from glucose. Leucine and isoleucine metabolism of fish meat led to increased 2−methyl−1−butanol, 3−methyl−1−butanol, 2−methylbutanal, and 3−methylbutanal; sulfur−containing volatiles are mainly generated by microbial−mediated enzymatic degradation of cysteine, methionine, and derivatives (e.g., DMS shown in Figure 3).

### 3.3. Living Environment

Due to increased consumption demand, aquaculture requires high stocking densities and feed supplies to satisfy productivity, which, however, lead to unfavorable eutrophication. Simultaneously, high fish stocking densities (>10,000/ha) and feeding rates (>70 kg/ha/d) fuel the rapid growth of algae, particularly cyanobacteria, which are known to produce undesirable flavors [72,73]. In addition, the relationship between the temperature and photoperiod associated with climate warming may affect phytoplankton growth [74]; in particular, the cyanobacteria growth rate and their ability to produce toxins are positively correlated with temperature elevation [75,76]. Rising temperature can result in cyanobacterial bloom enhancement, which poses a threat to water quality [77]. Fish are very susceptible to consuming food as well as industrial pollutants and natural off−odor compounds existing in their living environment [33]. These undesirable flavors from the environment render fish unmarketable unless purified by large quantities of clean water, causing a heavy economic burden on the aquaculture industry.

Moreover, it was shown that the concentrations of MIB, GSM, and β−cyclocitral are high in pond water, yielding undesirable fish flavors [18]. Furthermore, studies showed that MIB, GSM, β−cyclocitral, and dimethyl trisulfide are volatile compounds that frequently exist during cyanobacterial bloom episodes and were successfully detected in all predominant odor compositions from fish tissue, sediment, and algal cell samples [78,79,80,81]. Typically, cyanobacteria, certain fungi, and various actinomycetes produce these flavor metabolites and excrete them into the environment: Streptomyces can produce MIB and GSM, and *nannocystis* has been shown to produce MIB [82,83] (the biosynthesis of MIB and GSM is shown in Figure 4); Microcystis promotes β−cyclocitral synthesis [84]; microalgae, seaweed, and plankton are rich in a large quantity of dimethyl−β−propiothetin (DMPT, a precursor of DMS) [4]; and cyanobacteria and fish feed are the primary sources of odor−active terpenes [20,85,86]. Fish then take up these metabolites across their gill membranes, leading to the accumulation of these compounds within tissues that are rich in lipids. Besides the sources mentioned above, drinking water treatment plants and industrial waste treatment facilities are also key factors in producing undesirable flavors. For example, naphthalene compounds such as 2,6−dimethylnaphthalene and 2−methylnaphthalene, the degradation products of factory waste via microorganisms, can accumulate in fish as environmental pollutants [87]. Accidental spills of petroleum hydrocarbons also cause pollution in the same way [83].

## 4. Odor Control Techniques

Increased fish supplies are required to meet increased human consumption demands, while undesirable flavor contamination substantially delays harvest, thereby causing economic losses for fish farmers [83]. Effectively controlling the odor of fish products is critically important.

### 4.1. Environmental Renovation

Both GSM and MIB, derived from the culture environment, are the primary earthy and musty compounds associated with fish. The current method is to move the fish to a large body of clean water and stop feeding in order to purge the undesirable flavor compounds from the fish’s tissue [88]. However, this approach may take days or weeks to obtain the lowest residual levels of geosmin and MIB in the fish flesh, depending on various factors, such as the intensity of the undesirable flavor, the water temperature, and the fat content of the fish flesh [15,89,90]. In addition, fasting will greatly reduce the quality of fish [91]. In addition to temporary cultivation with clean water, strategies such as hindering the growth of bacteria and algae that produce odors and the adsorption or removal of undesirable flavor substances in aquaculture water are usually adopted.

The results obtained in previous studies demonstrate that aerobic, organic−rich conditions are beneficial to the growth of bacteria [88], and certain nutritional factors can stimulate GSM production by actinomycetes [15]. As a result, the biofloc technology (BFT) production system was launched to maintain the water’s turbulence through continuous aeration, make bacteria lose their cell buoyancy regulation ability, and metabolize excreted feed nitrogen, thus decreasing cyanobacteria and actinomycetes and significantly weakening the intensity of undesirable “earthy” and “musty” flavors [92].

In order to solve the undesirable flavor problem of cyanobacteria, accumulating experimental evidence has proven that ultrasound technologies have also shown significant potential in the management of cyanobacteria and possess the advantages of energy conservation, safety, and cleanness [93,94,95]. One of the damage mechanisms owing to ultrasonic irradiation is the increased presence of free radicals that destroy cellular constituents and functions by inducing lipid peroxidation, damaging cellular membranes, and inhibiting photosynthesis [96]. The relative content of malondialdehyde (MDA), used as a quantitative indicator of lipid peroxidation, was significantly increased after ultrasonic irradiation in most species of cyanobacteria [97]. In addition to algae removal, ultrasonically induced cavitation has been demonstrated to directly reduce off−flavor compounds GSM and MIB from RAS water, and high−frequency ultrasound (850 kHz) was more effective compared to low−frequency ultrasound (20 kHz) [98].

Due to the hydrophobic structures of GSM and MIB, adsorbents such as activated carbon and zeolite are often used to adsorb and remove odorous substances in water, which have noticeable effects [99]. However, natural organic substances in ponds are also adsorbed, thus greatly reducing the adsorption capacity of activated carbon [100]. Ozone (O_3_) is a kind of oxidant used in water treatment because of its high oxidation potential. Under specific conditions, O_3_ catalyzes the decomposition of hydroxyl radicals to form highly oxidizing hydroxyl radicals and then oxidizes GSM and MIB [101]. However, Atasi et al. reported that conventional ozonation degradation of GSM and MIB has a strong dependence on the dose of ozone: when the dose reaches 8 mg/L, the removal rate is still less than 30%, and a high dose of ozone may cause toxic side effects on aquaculture [102,103]. There are other oxidation treatments for GSM and MIB, including ultraviolet (UV) radiation and advanced oxidation processes using different catalysts, and the combination of different treatments, such as UV−TiO_2_ photocatalysis [104] and the combined use of O_3_ and UV [105], can greatly improve the degradation rate.

A biological control method involves the introduction of specific microorganisms into the aquaculture environment to reduce the odor substances in water and fish through microbial metabolism. Compared with physical and chemical methods, this method is more environmentally friendly. To date, *Bacillus* spp. [106], *Stenotro−phomonas* spp. [107], *Pseudomonas* spp. [108], *Enterobacter* spp. [109], *Micrococcus* spp. [110], *Flavobacterium* spp. [111], and *Brevibacterium* spp. [110] have been found to be able to use GSM and MIB for normal metabolism. Although MIB and GSM can be removed by microbial degradation, the degradation rate is quite slow [108,110]. Biodegradation combined with photocatalysis has been proved to have the potential to repair natural water contaminated by odorous substances [112,113]. Fu et al. [114] developed a tightly coupled photocatalytic and biodegradation system to remove GSM and MIB, which significantly improved the removal efficiency of MIB and GSM.

### 4.2. Processing Treatment 

#### 4.2.1. Freezing

Decreasing the storage temperature is a common and natural preservation method used to increase the stability of fish and commercial fish products. Rahman et al. found that with an increase in storage temperature, the rate of lipid oxidation in dried grouper increases significantly [115]. It has been shown that −35 °C is the optimal temperature for maintaining high−quality fish for long−term storage [116]. Ultra−low−temperature storage (<−40 °C) can inhibit biochemical reactions, but the use of ultra−low−temperature storage has a negative impact on the structure of fish due to ice crystal generation and increases the possibility of fish tissue rupture and costs. Storage at −35 °C with oxygen barrier material packaging is sufficient to stabilize proteins, inhibit the formation of TMAO in fish tissue, and maintain good texture characteristics of lean fish. In addition, on the basis of a suitable temperature, increasing the freezing speed and reducing the size of ice can effectively reduce damage by ice crystals [117]. Furthermore, increasing the freezing rate also reduces the destructive effect of ice crystals, which effectively diminishes the size of the ice. The results showed that carp (*Cyprinus carpio*) samples treated with ultrasound−assisted immersion freezing (UIF) at 180W ultrasonic power reduced the freezing time compared to the control groups with no ultrasound treatment. Meanwhile, UIF was shown to retard the growth of TBARS and total volatile basic nitrogen values (TVB−N) when compared to air freezing (AF) and immersion freezing (IF) during storage [118].

#### 4.2.2. Salting and Drying

Salting and drying have been applied to suppress the growth of Gram−negative bacteria and inhibit enzyme−related chemical reactions in meat products by reducing water activity [119]. In traditional manufacturing, fish fillets, fish pieces, or whole fish are exposed to a fluidized bed full of salt particles in hot and dried air, controlling the time and temperature of drying and salting. Drying produces a unique flavor of products, catering to the demands of consumers. There are also several different drying methods, such as natural sun, hot air, vacuum solar, electric heat, and microwave drying. Studies have indicated that sensory indices and the chemical composition of salted fish are not associated with different drying methods. However, the microwave drying efficiency outperforms that of other drying methods in terms of microbial bacteria [120,121].

#### 4.2.3. High−Pressure Processing

High−pressure processing (HPP) can change cell morphology and damage major components, such as bacterial cell membranes and walls, as well as several organelles, reducing microbial loads within meat and seafood products. Meat products receive uniform treatment under high pressure and maintain high sensory quality due to its low impact on flavor [122]. It has been shown that HPP increases the hardness and springiness of frozen hake. Moreover, cooked hake is also influenced by HPP and obtains the best quality at 300 MPa during 6 months of frozen storage [123]. Meanwhile, HPP inactivates the endogenous cathepsin that easily causes the deterioration of chilled fillets; for instance, bluefish cathepsin C nearly lost its activity after treatment at 300 MPa for 30 min, providing better fish quality [124].

#### 4.2.4. Boiling

Boiling can suppress lipid deterioration in fish in several ways, such as denaturing lipoxygenases, forming water−soluble antioxidants, and destroying heme compounds. As a result of heat treatment, the quantity of aldehydes in fish has been shown to drop to nearly undetectable levels due to carbonyl−amino reactions [125]. Meanwhile, Kim et al. indicated that the fish gelatin of dried anchovies was hydrolyzed after boiling, forming an invisible edible film that protected against oxidative rancidity [126]. Boiling could slow the lipid hydrolysis process of dried sardines but adversely led to the loss of PUFAs, ultimately damaging sensory characteristics during storage [127].

#### 4.2.5. Fermenting

Bacteria as a starter have significant potential in improving the flavor of fish products. *T. halophilus* is used in the fermentation of fish sauce, which significantly improves the amino acid composition of the product and reduces the concentrations of undesirable flavor compounds, such as dimethyl disulfide and biogenic amines [128]. Similarly, in Thailand, *Staphylococcus xylosus* is used to produce fish sauce in order to change the flavor notes. As a result, the sensory evaluation indicated that the fishy, fecal, rancid, and sweaty notes of fish sauce inoculated with the bacterium were weaker than those of the fish sauce without treatment [129]. In addition, irradiation−assisted salting and fermentation can significantly improve sensory flavor characteristics, especially by reducing the typical fishy smell and improving color and microbial safety [130].

#### 4.2.6. Defatting

Removing the fat from fish is a direct method used to inhibit unpleasant odors caused by lipid oxidation. Enzymes, organic solvents, and alkaline treatment can be used to achieve the effect of degreasing. For example, lipoxygenase extracted from marine macroalga reduced the undesirable odors of fish oil by site−specific cleavage of hydroperoxides, producing more desirable alcohols, aldehydes, and ketones, thus releasing fresh fish and fruit flavors [131]. The gelatins from seabass skin use citric acid and isopropanol alcohol to remove lipids, inhibiting the abundance of volatile compounds, thereby lowering fishy odors, the peroxide value, and TBARS [132]. The protein isolate separated by acid or alkaline solubilization and isoelectric precipitation from Nile tilapia and broad−head catfish had lower GSM and MIB concentrations as well as a negligible muddy odor [133,134]. In this regard, using a polar antioxidant can effectively prevent oxidation in protein isolates regardless of pH treatment [34].

#### 4.2.7. Masking

Seasonings can be used as masking agents that are directly added to food during the cooking process to cover up unpleasant odors. Catfish fillets blended with lemon pepper and soaked in a food container satisfied the majority of evaluation panelists. Due to the presence of masking agents, the muddy/earthy odorant MIB was perceived less or even not at all [135]. Washed minced fish were treated with piper guineense and salt to make kamaboko. The addition of piper guineense increased the kamaboko score for taste and overall acceptability, as well as significantly reduced the microbial flora of kamaboko [136]. In India, Japan, China, and Southeast Asia, ginger is very suitable for fish dishes, bringing sensations of pungency and hotness to mask undesirable flavors. In addition to ginger, cumin, coriander, basil, mint, and celery are often used in Asian cuisine recipes [137].

### 4.3. Application of Additives

#### 4.3.1. Synthetic Additives

Legal food additives with specified contents are indispensable in the industrial production of fish products. A food additive, which can be synthetic or natural, is normally not consumed as a food itself but is intentionally added to food to improve aroma, taste, texture, or shelf−life [138]. As undesirable flavors are primarily caused by oxidation, it is beneficial to add antioxidants to food systems in order to reduce oxidation. Antioxidants, such as propyl gallate (PG), butylated hydroxytoluene (BHT), butylated hydroxyanisole (BHA), tert−butylhydroxyquinone (TBHQ), vitamin E (tocopherol), vitamin C (ascorbic acid), phosphates, and citrate, have been used individually or in combination to suppress the oxidation process [139]. The addition of antimicrobial agents is also vital in inhibiting the development of microorganisms, thereby improving the appearance and flavor of fish products. Lactic, sorbic, and benzoic acids and their salts can be extracted directly or obtained synthetically and are effective organic compounds widely used as antimicrobial agents to prolong the sensory quality [140]. Nitrite is an ordinary synthetic antimicrobial compound that is used to control bacteria and fungi during meat preservation, such as *Flavobacterium*, *Micrococcus*, and *Pseudomonas*; however, its dosage is strictly controlled to prevent harm to the human body at high dosages [141]. Meanwhile, in industrial production, seasonings, antioxidants, and antibacterial agents (or preservatives) can be added to the product formulation according to regulations (e.g., in the United States, the additive must be GRAS−listed (Generally Recognized as Safe) according to the American Food and Drug Administration (USFDA, 2009). In Canada, it must fall under GMP (Good Manufacturing Practice) in accordance with the Canadian Food and Drug Act (HC, 2006). In China, it must comply with “National Food Safety Standards for Food Additives” (GB 2760−2014)). These active solutions can be used to spray and penetrate the raw fish product in order to control the odor [140].

#### 4.3.2. Natural Additives

Great attention has been paid to natural additives because synthetic additives have been implicated as potentially toxic and carcinogenic hazards in past decades. With the overall improvement of living standards, natural materials with antioxidant activity became more popular for consumers. Immersion or coating is the conventional treatment for applying natural antioxidants to fish or other seafood. Clove water extract is applied to oven−dried omena fish (*Rastrineobola argentea*) by immersion, significantly reducing the concentrations of TBARS and peroxide values of omena fish [142]. Moreover, caffeic acid is an active antioxidant that prevents lipid oxidation in the minced white muscle of horse mackerel during frozen storage [143]. Furthermore, grape polyphenols can inhibit lipid oxidation in frozen minced fish muscle, protecting the endogenous antioxidant system of fatty fish [144,145]. Similarly, tea polyphenols effectively inhibit TMAO breakdown and the oxidation of lipids and therefore maintain the quality of dried−seasoned squid [146]. Tan and Shahidi demonstrated that phytosterol displayed an excellent antioxidant effect as well [147]. In general, active substances function as antioxidants through several action mechanisms, such as chelating prooxidative metal ions [148], scavenging free radicals, suppressing oxidative enzymes and reactive oxygen species, or interacting with bio−membranes [149].

Moreover, besides lipid oxidation, phenolic compounds can play a significant role in preventing microbial spoilage; for instance, catechin has been shown to have great antimicrobial activity against bacteria that produce H_2_S in fish and fish products [150]. Tannic acid was also reported to retard the growth of psychrophilic bacteria and inhibit the increase in the total viable count in striped catfish slices during refrigerated storage [151]. Citrus essential oil was shown to inhibit the growth of pathogenic and fungi flora in sea bass fillet [152]. Oregano (0.8%) [153], thyme (1%), and laurel essential oils (1%) [154] were also shown to improve the quality of fish. Interestingly, many studies have reported that essential oils contain phenolic compounds, such as eugenol, thymol, or carvacrol [155]. These phenolic compounds can lyse the cell walls of microorganisms, disrupt membrane proteins, further damage various enzymatic systems, and inactivate genetic material in order to strengthen their antimicrobial properties [156]. Recent research has also indicated that phenolic compounds can be chemically reactive with various food constituents, such as proteins, or directly react with volatile odor compounds to modify the product’s flavor [157,158]. Furthermore, bacteriocins, namely, small bacterial peptides, showed strong antimicrobial activity. Nisin has also been used to control the quality of snakehead fish fillets during cold storage [159].

Natural products, especially antioxidants and antimicrobial agents of natural origin, should be widely studied as safe alternatives to synthetic additives. Moreover, different plants and microbes should be screened qualitatively and quantitatively for the presence of potent active compounds and their potential uses in fish and fish products.

### 4.4. Packaging 

#### 4.4.1. Vacuum Packaging (VP)

With the decreasing usage of synthetic additives, the packaging, used as the last barrier, becomes more and more important before the distribution and storage of fish products. Preventing fishery products from coming into contact with oxygen is a precautionary measure against oxidative deterioration. Vacuum packaging (VP) and modified atmosphere packaging (MAP) of meat and meat products have gained importance in improving shelf−life and avoiding the development of rancidity [160]. In VP, there are no air gaps between the product and the packaging. Moreover, VP is used for packaging frozen fish or preserved fish products in order to prevent the formation of undesirable flavors from oxidation. However, due to the enhancement of trimethylamine formation under anaerobic conditions, VP is not recommended to apply to marine fish products [161]. 

#### 4.4.2. Modified Atmosphere Packaging (MAP)

Carbon dioxide (CO_2_), nitrogen (N_2_), and oxygen (O_2_) are the principal gases in MAP, with each having different purposes and functions. Typically, CO_2_ is used to inhibit bacteria and mold, N_2_ is used to prevent lipid oxidation and package collapse, and O_2_ is used to prevent the growth of anaerobes [162]. In many fishery products, MAP with high CO_2_ has been proven to be more effective than VP in inhibiting the growth of spoilage microorganisms. CO_2_ can penetrate bacterial cytomembranes and influence cytoplasmic enzyme activity. Meanwhile, it is critical to control the storage temperature for MAP. Higher temperatures result in the reduction of dissolved CO_2_ within the product, leading to higher microbial and enzymatic activity and consequently damaging the quality of the product [163]. In addition, recent studies have indicated that super−chilling prior to MAP is a valuable measure that has a significant impact on bacterial inhibition. MAP super−chilled fillets were shown to have a longer shelf−life and lower bacterial counts compared to other chilled fillets [164].

#### 4.4.3. Active Packaging (AP)

Active packaging (AP) techniques are concerned with substances that absorb oxygen, ethylene, moisture, CO_2_, and flavors/odors and those that release CO_2_, antimicrobial agents, antioxidants, and flavors [165]. In general, packaging materials need proper water and gas barrier capabilities as well as excellent sealing properties. In practical applications, polyethylene (PE) and polypropylene (PP) have excellent water barrier properties [166]. Ethylene vinyl alcohol (EVOH) exhibits excellent gas barrier properties [167]. These are synthetic polymers widely used in food packaging, but they cannot undergo physical, chemical, or biological degradation and have thus caused numerous severe environmental and health−related problems [168,169]. Therefore, there is an increasing interest in the development of environmentally friendly biodegradable polymers (i.e., biopolymers) for packaging materials. Biopolymers based on polysaccharides, proteins, and lipids from numerous plant and animal sources can be formed into either edible films or coatings and have suitable application properties [170,171]. Chitosan is one of the most popular polysaccharide biopolymers that can form a semipermeable film with intrinsic antimicrobial activity [172]. Blending active components in packaging material can also improve the barrier characteristics. For example, chitosan blended with various antimicrobial agents, such as tea tree essential oils and cinnamon oil, was made into an improved antimicrobial film to enhance the odor, texture, and color of trout fillets [173,174]. For wrapping dried anchovy, chitosan film containing acetic or propionic acid was shown to have a superior effect on oxidative stability compared to polyester–polyethylene laminate during five months of storage [175]. A growing trend of packaging is to integrate water absorbers, oxygen scavengers, and antimicrobial agents into the packaging material rather than apply them as individual sachets [176,177].

## 5. Conclusions

In recent years, in the pursuit of health, fish have been favored by consumers because they are rich in high−quality proteins and PUFAs. However, undesirable flavors limit consumers’ purchase and consumption. The undesirable flavors of fish are mainly due to the water quality in the aquaculture environment and deterioration reactions (enzymatic reactions, lipid autoxidation, and microbial actions). The synergistic effect of carbonyl compounds, alcohols, GSM, MIB, TMA, and other substances gives fish a worse flavor. In order to develop high−value fish products, odor removal from fish during production, processing, transportation, and consumption has been widely studied. Traditional methods, such as basic aquaculture management, salting, freezing, masking, and heat treatment, have been widely used in fish production and processing, but many of them require further improvement. More deodorization technologies for fish have been deeply explored, such as functional microbial degradation, ultrasonic irradiation, the addition of natural antioxidant and antimicrobial agents, and fresh−keeping packaging. Among them, due to the synergistic effect, the combined use of two or more deodorization strategies usually shows more effective results. At present, existing deodorization technologies and methods are diverse, but they have certain limitations and limited scopes of application. To sum up, it is necessary to establish a comprehensive, applicable, and efficient deodorization scheme that can satisfy consumers’ demand for better sensory quality of fish and fish products.

## Figures and Tables

**Figure 1 foods-11-02504-f001:**
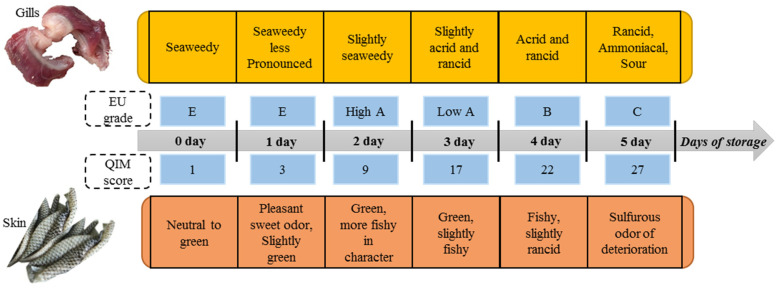
Freshness assessment of iced−stored sardine with emphasis on odor development according to EU grading and QIM (the EU grade is the European Union grade; the QIM score is the Quality Index Method score). The EU freshness grading distinguishes four categories of fish, from E (very fresh state), A, and B to C (not admitted). The QIM uses many weighted parameters (e.g., appearance, eyes, cover, and gills) with a scoring system from 0 to 4 demerit points for each parameter; it gives a total score of zero to very fresh fish and returns an increasingly larger result as fish deteriorates [26].

**Figure 2 foods-11-02504-f002:**
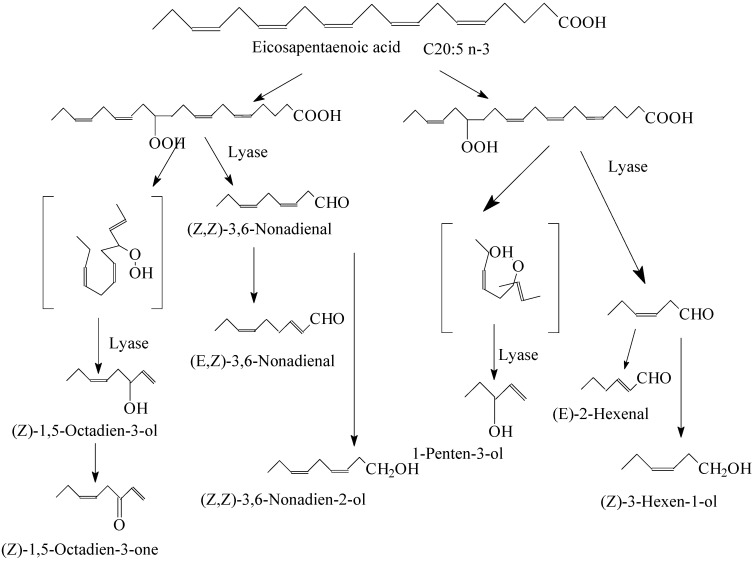
Proposed mechanism for biochemical reactions of eicosapentaenoic acid.

**Figure 3 foods-11-02504-f003:**
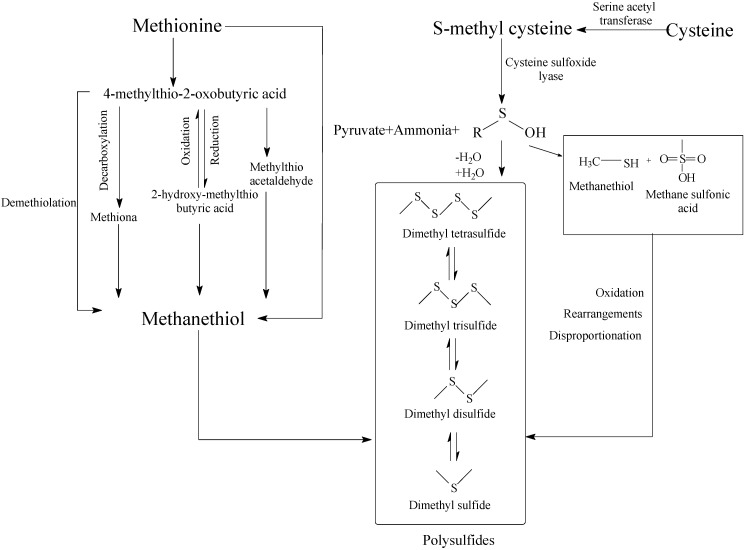
Enzymatic degradation of cysteine and methionine, generating DMS.

**Figure 4 foods-11-02504-f004:**
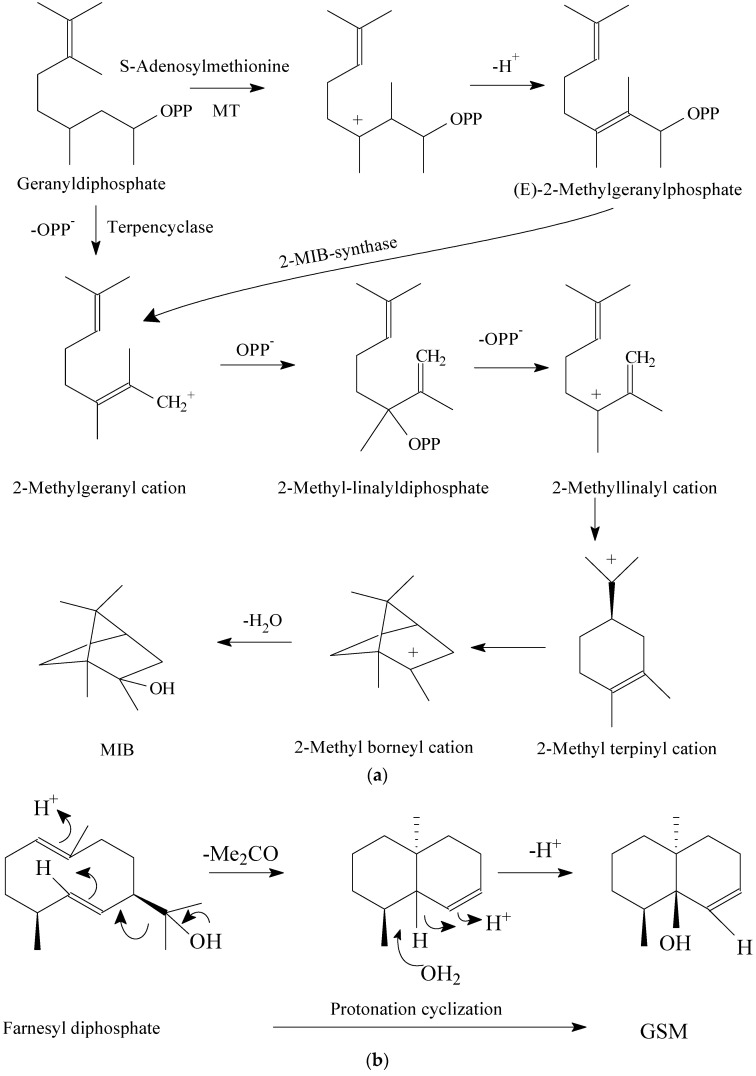
(**a**) Biosynthesis of MIB via methylation of geranyldiphosphate (GPP) and cyclization of (E) −−2−−geranyldiphosphate; (**b**) biosynthesis of GSM via 1,2−−hydride shift, loss of hydroxypropyl moiety, and capture of water.

**Table 1 foods-11-02504-t001:** The main volatiles derived from lipids contributing to off−flavors.

No.	Off−Flavors	Origin	Oxidation Causes	Refs.
1	1−Penten−3−ol	Eicosapentaenoic acid	15−Lipoxygenase	[13]
2	(E)−2−Pentenal	Linolenic acid, docosahexaenoic acid /*n*−3 polyunsaturated fatty acids	15−Lipoxygenase	[42]
3	Hexanal	Linoleic acid /*n*−6 Polyunsaturated fatty acids	15−Lipoxygenase/autoxidation	[43,44]
4	(E)−3−Hexen−1−ol	Eicosapentaenoic acid	15−Lipoxygenase	[13]
5	(E)−2−Hexenal	Linolenic acid /*n*−3 polyunsaturated fatty acids	15−Lipoxygenase	[13,42]
6	Heptanal	*n*−6 Polyunsaturated fatty acids	Autoxidation	[42,43,44]
7	1−Octen−3−ol	Arachidonic acid, linoleic acid /*n*−6 polyunsaturated fatty acids	12−Lipoxygenase	[42,44]
8	(Z)−1,5−Octadien−3−one	Eicosapentaenoic acid /*n*−3 polyunsaturated fatty acids	12−Lipoxygenase	[13,43]
9	Nonanal	*n*−9 Polyunsaturated fatty acids	12−Lipoxygenase	[42,43]
10	(E)−2−Nonenal	Linoleic acid, arachidonic acid	12−Lipoxygenase	[43]
11	(E,Z)−2,6−Nonadienal	Eicosapentaenoic acid /*n*−3 polyunsaturated fatty acids	12−Lipoxygenase	[13,43]
12	2,4−Heptadienal (two isomers)	Linolenic acid /*n*−3 polyunsaturated fatty acids	12−Lipoxygenase/autoxidation	[43]
13	2,4−Decadienal (two isomers)	Linoleic acid	Autoxidation	[45]
14	Short− and branched−chain fatty acids (e.g., butanoic, 2−/3−methylbutanoic, hexanoic, and octanoic acids)	Fatty acids	Autoxidation	[46]

**Table 2 foods-11-02504-t002:** Specific spoilage microflora dominating in fresh fish meat during cold storage under different gas atmospheres.

Gas Composition	Microflora
Air	*S. putrefaciens*, *Pseudomonas* spp.
>50% CO_2_ with O_2_	*B. thermosphacta*, *S. putrefaciens*
50% CO_2_	*P. phosphoreum*, *Lactic acid bacteria*
50% CO_2_ with O_2_	*P. phosphoreum*, *Lactic acid bacteria*, *B. thermosphacta*
100% CO_2_	*Lactic acid bacteria*
Vacuum packaged	*Pseudomonas* spp.

From Reference [57].

**Table 3 foods-11-02504-t003:** VOCs that common bacteria (e.g., *Pseudomonas* spp. and *Shewanella* spp.) produce in fish during aerobic storage and their precursors and attributes.

Compounds	*Pseudomonas*	*Shewanella*	*Lactic Acid Bacteria (LAB)*	Precursor(s)	Flavor Descriptors	Refs.
**Alcohols**						
2−Methyl−1−butanol	Y	Y	/	Isoleucine	Malt, wine, onion	[63,64]
3−Methyl−1−butanol	Y	Y	Y	Leucine	Whiskey, malty, burnt	[63,64]
Ethanol	Y	Y	Y	Glucose	Alcoholic, ethereal, medical	[63,65]
**Aldehydes**						
2−Methylbutanal	/	/	Y	Isoleucine	Cocoa, coffee, fruit	[63,66]
3−Methylbutanal	/	/	Y	Leucine	Sweet, malty, sour	[63,66]
Benzene acetaldehyde	/	/	Y	Phenylalanine	Sweet, honey sweet	[67]
**Ketones**						
3−Hydroxy−2−butanone	/	/	Y	Glucose	Butter, creamy, dairy, milk, fatty	[63,68]
2−Heptanone	Y	Y	/	Fatty acid	Fruity, spicy	[63,69]
**Esters**						
Ethyl acetate	NAD	/	Y	Multiple origins	Ethereal, fruit, sweet	[68]
Ethyl octanoate	Y	/	NAD	Multiple origins	fruit, fat	[63,70]
3−Methylbutyl acetate	/	/	Y	Multiple origins	Fruit, sweet, banana, ripe	[63,69]
**Organic acids**						
Acetic acid	/	Y	Y	Glucose	Pungent sour	[55,63,69]
**Sulfur compounds**						
Hydrogen sulfide	/	Y	Y	Cystine, cysteine, methionine	Rotten eggs	[23,69]
Methanethiol	Y	Y	/	Methionine, cysteine	Sulfur, gasoline, garlic	[23,49,64]
Dimethyl sulfide	Y	Y	/	Methanethiol, methionine, cysteine	Cabbage, sulfur, gasoline	[23,49]
Dimethyl disulfide	Y	Y	/	Methionine, cysteine	Onion, cabbage, putrid	[23,63,64]
Dimethyl trisulfide	Y	Y	/	Methionine, methanethiol, cysteine	Sulfur, fish, cabbage	[23,69]
**Nitrogen compounds**						
Ammonia	NAD	NAD	NAD	Amino acids (e.g., arginine, histidine, tyrosine)	Ammoniacal	[71]
Trimethylamine	/	Y	/	Trimethylamine oxide	Fishy, oily, rancid, sweaty	[49,68,71]

NAD, no available data; Y, can produce; /, cannot produce. Flavor descriptors according to: Flavornet (http://www.flavornet.org/flavornet.html, accessed on 13 June 2022); The Good Scents Company (http://www.thegoodscentscompany.com/, accessed on 13 June 2022); The kinds of volatile compounds are in a bold.

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
