# Peer review of "Advances in the Formation and Control Methods of Undesirable Flavors in Fish"

_foods, 2022, doi:10.3390/foods11162504_

Round 1
Reviewer 1 Report
Line 48: Statement “which was consistent with the fact that the oily fish often have more volatile oxides than lean fish” need to support with suitable reference(s).
Line 49: Current suitable term for organoleptic evaluation is sensory evaluation. Please update to Sensory differences
Line 119: Need to elaborate in detail the storage effects and its relationship with the volatile compounds as well as for living environment; microbial metabolites and lipid oxidation.
Line 308: Need additional information on Active Packaging (AP): Better divide sub-sub chapter 4.4 Packaging to sub-sub-sub chapter 4.4.1. Vacuum Packaging (VP); 4.4.2. Modified Atmosphere Packaging (MAP) and 4.4.3. Active Packaging (AP).
Reviewer 2 Report
Line 47: i recommend delete (p < 0.05)
Line 64: check quote
Additionally, i suggest others tables as table 3 to Processing treatment and Application of additive.
Reviewer 3 Report
This MS reviews the formation of undesirable flavors in fish and its control methods. Overall, the topic is interesting, and this MS is well-written and informative. However, it is still seems like a book chapter providing only information but lack of discussion. Thus, It would be great if there would be more insights or the authors’ perspectives.
My suggestion for improvement this review are as follows:
- Figure1 is difficult to understand and read. Mainly, it lacks of the references related with this information. Moreover, the structure/pattern is confused i.e. how is the relationship between the same color??? What exactly compounds related with each flavor characteristics such as mild green metallic??? etc. Moreover, the full name of each compounds is need (only abbreviation is not proper). Please re-construct the figure for better understanding.
- The discussion related with figure 2 is too less informative. It is better if author describe why gills and skin were the selected part related with undesirable flavor in fish (why other part is not concern). How’s volatiles related with each flavor and how’s it develop as extended the storage time. Moreover, the acceptability level of bot EU and QIM standard should be provide to gain more information.
- Please provide the reference in table 2
